# Clustering of Monolingual Embedding Spaces

Kowshik Bhowmik [1,*] and Anca Ralescu [2]

1 The College of Wooster, Mathematical and Computational Sciences, Wooster, OH 44691, USA
2 Electrical Engineering and Computer Science, University of Cincinnati, Cincinnati, OH 45221, USA
* Correspondence: kbhowmik@wooster.edu

**Abstract:** Suboptimal performance of cross-lingual word embeddings for distant and low-resource languages calls into question the isomorphic assumption integral to the mapping-based methods of obtaining such embeddings. This paper investigates the comparative impact of typological relationship and corpus size on the isomorphism between monolingual embedding spaces. To that end, two clustering algorithms were applied to three sets of pairwise degrees of isomorphisms. It is also the goal of the paper to determine the combination of the isomorphism measure and clustering algorithm that best captures the typological relationship among the chosen set of languages. Of the three measures investigated, Relational Similarity seemed to capture best the typological information of the languages encoded in their respective embedding spaces. These language clusters can help us identify, without any pre-existing knowledge about the real-world linguistic relationships shared among a group of languages, the related higher-resource languages of low-resource languages. The presence of such languages in the cross-lingual embedding space can help improve the performance of low-resource languages in a cross-lingual embedding space.

**Keywords:** cross-lingual word embeddings; low-resource languages; bilingual lexicon induction; degree of isomorphism

## 1. Introduction

Mapping-based methods of inducing cross-lingual word embeddings are based on the assumption that semantic concepts are language-independent [1]. This assumption led to learning an orthogonal or isomorphic map from one monolingual embedding space to another using known translation word pairs between two languages. Cross-lingual embedding space induced in this manner can enable tasks such as Bilingual Lexicon Induction and Machine Translation and can also lead to the transferring of knowledge from one language to another. However, the reported performance of such tasks showed suboptimal performance for low-resource languages, which are also often languages that are typologically distant from English and other resource-rich European languages [2]. These results show the weakness of the isomorphic assumption. Henceforth, researchers have proposed several explanations for the varying degrees of isomorphism between independently trained monolingual embedding spaces, notable among which are typological differences among the languages in question and the comparative resources on which the word embeddings were trained [3]. To investigate the comparative impact of these two factors, this research employs two clustering algorithms: Hierarchical and Fuzzy C-Means on pairwise similarity/distance values computed among the chosen set of languages. The languages are diverse both in terms of the language families they belong to and the amount of available resources. Three measures of isomorphism reported in the literature were utilized: Eigensimilarity [1], Gromov–Hausdorff distance [4], and Relational Similarity [3]. Another aim of this research was to determine the combination of the measure of isomorphism and clustering algorithms that best aligns with our existing knowledge of the language families. This was performed with the view to finding out the measure of isomorphism that can substitute for a measure of linguistic similarities among a group of languages. The

presence of related higher-resource languages has been shown to improve the performance of low-resource languages in the cross-lingual embedding space [5]. Being able to cluster typologically similar languages together will lead to a better representation of low-resource languages in the cross-lingual embedding space.

## 2. Materials and Methods

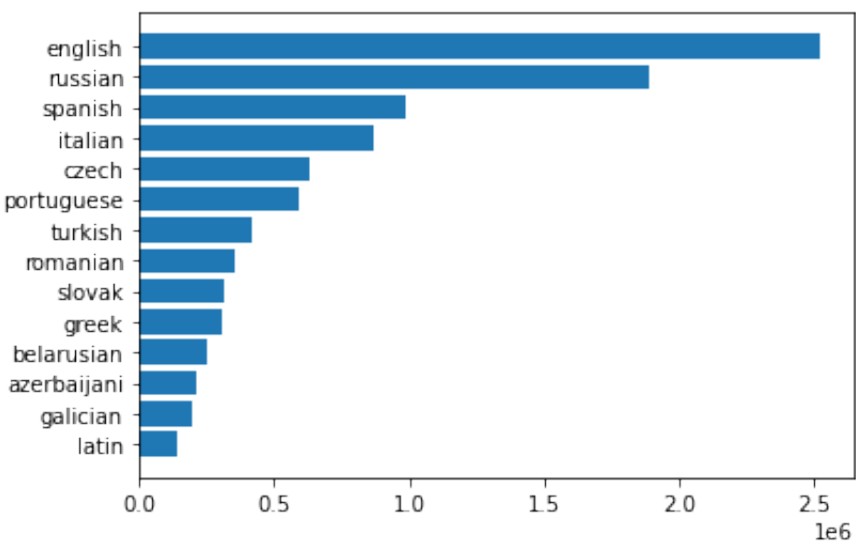

**Figure 1.** Number of Words (in millions) in Monolingual Embedding Spaces.

### 2.1. FastText Word Vectors

Popular word embedding models, such as the Skip-gram and Continuous Bag-of-Words (CBOW), assign each word in their vocabulary a distinct vector. Such methods ignore the internal structure of words. This is a serious limitation for morphologically rich languages, such as Turkish and Finnish. Such languages also contain rare words in their vocabulary, good vector representations for which they are difficult to learn. The approach proposed by Bojanowski et al. [6] extends the popular skip-gram model to represent each word as a bag of character n-grams. The words are then represented as the sum of their character n-grams. They propose a scoring function in order to encode information about the internal structure of words. Boundary symbols are used to distinguish between prefixes and suffixes from the rest of the character n-grams making up a word. The original word is also a part of its character n-grams in order to distinguish identically spelled n-grams belonging to different words. If there is a word *midnight* and $n = 3$, the word is represented by the following n-grams:

$$< mi, mid, idn, dni, nig, igh, ght, ht >$$

along with the special sequence
$$< midnight > .$$

The sum of the n-gram vectors is used to represent the word. Let $G_w \subset \{1, ..., G\}$ be the set of n-grams of word $w$ in the vocabulary. The scoring function *s(w, c)* is obtained as:

$$s(w, c) = \sum_{g \in G_w} z_g^T v_c.$$

Here, $z_g$ is the vector representation of the character n-gram $g$. Such a representation allows the learning of compositional and rare words by sharing the representation of the n-grams.

### 2.1.1. Embedding Size

Word vectors were trained using fastText on Common Crawl and Wikipedia. The CBOW model was used with position weights. These word vectors are 300 in dimension. They were trained on character n-grams that have a length of five, with the window size being five, along with ten negatives.

The disparity of resources among the different embedding spaces in our chosen language set can be seen in Figure 1. For the purpose of our research, the monolingual embeddings in our language set are divided into three groups based on their relative resources. English, Russian, Spanish, Italian, Portuguese, and Czech have more than 500,000 words in their embedding spaces; they will be categorized as high-resource languages. However, it should be mentioned that the English space is made up of around 2.5 million words, while the Russian space has 1.8 million words in it. The Greek, Slovak, Romanian, and Turkish space range from 300,000 to 500,000 words. They will be referred to as moderate-resource languages. Latin, Galician, Azerbaijani, and Belarusian will be referred to as low-resource languages. They have less than 300,000 words in their embedding space. This classification is based on their respective Wikipedia sizes and not the number of people who speak these languages as a first language.

### 2.1.2. Language Families

The foundational hypothesis for the research on the relatedness of languages states that the shared features between two languages can be attributed to their common ancestor [7]. The family-tree model of languages followed by linguists is a hierarchical structure that captures the divergence of the languages from their ancestors and the level of similarity each language shares with the rest of the languages in the tree. Although integral to work in historical and comparative linguistics, this family-tree model has some issues. Genetic relatedness contributes to the shared features between languages, but there are other possible factors that can account for them. First, it is possible for two languages to share some features purely by chance. Some of the features shared between two languages may actually be common to all languages or, in other words, universal language manifestations. Finally, two languages may have features in common because of borrowing. One of these languages may have borrowed aspects of the other, or both of them may have borrowed it from a third language. For example, the large number of loan words in English from French may lead someone to believe that English is a Romance language instead of a Germanic one. One of the most difficult challenges faced by linguists is distinguishing between cognates resulting from forms going back to a common ancestor and those from borrowing.

English is part of the West-Germanic part of the larger Indo-European language family [8]. The most widespread language in the current world, English, boasts a cosmopolitan vocabulary. English has extensively borrowed from German and other Germanic languages as well as from Latin and French. Recent borrowings have taken the number of donor languages for English over 75.

Latin is the major language of the Italic branch of the Indo-European family of languages [9]. Linguists believe that with the spread of Roman power, Vulgar Latin acquired the dialectical variations from which the later Romance languages took shape. Latin survived long after the language itself stopped changing. As a medium for liturgy and learned discourse, it maintained homogeneity while the divergent Romance languages flourished.

Spanish is the most widely spoken Romance language. Spanish borrowed from Latin since the Renaissance and, more recently, from American English. Genetically, it belongs to the Iberian branch of the Romance language family along with Portuguese, Galician, and Catalan.

As mentioned above, Portuguese is an Iberian Romance language, a descendant of Vulgar Latin. Galician and Portuguese are closely related [10]. For a long stretch in the history of these two languages, the divergence between them was insignificant enough that they could be considered variants of the same language.

Romanian belongs to the Eastern or Balkan branch of the Romance language family [11]. Romanian has borrowed heavily from the Slavonic languages, and yet its Romance structure is left mostly unaffected. Many of these borrowed words are part of the religious vocabulary. However, some of these Slavonic words have their origin in Greek, hence making the influence of Greek on the Balkan Romance more significant than on the Western ones. Romanian also borrowed Greek words through the Latin language. A significant amount of Turkish words also made their way into the Romanian vocabulary, although some of them are no longer in practice. In recent centuries, Western Romance languages have seen increased influence on the Romanian language as well as English. Some of the English words borrowed by Romanian have their origin in Romance languages or Latin, making the situation more complex and interesting for linguists.

Russian and Belarusian, along with Ukrainian, are part of the Eastern branch of the Slavonic language family [12]. These languages have a high degree of mutual intelligibility.

Czech and Slovak are Western Slavonic languages that share plenty of similarities despite the two languages going through natural divergence processes. Apart from geography, geopolitics, and coming under the influence of different neighboring languages (German for Czech and Hungarian for Slovak), the two languages still hold mutual intelligibility of about ninety percent.

Turkish and Azerbaijani belong to the Turkic language family [13]. The languages in the family are very close to each other, with mutually intelligible dialects of different languages chaining them together, thus making it difficult to find clear boundaries among them.

The Greek language is the only language in its branch of the Indo-European family of languages [14]. Linguists focus on the Balkan branch of Greek to better understand its development in the periods of Middle and Modern Greek. The later stages of the language reveal features in Greek that are also present in other languages, such as Romanian, Aromanian, Albanian, etc.

### 2.2. Word Level Alignment of Cross-Lingual Word Embeddings

Mapping-based approaches of inducing a cross-lingual embedding space start by independently training monolingual word representations from large monolingual corpora [15]. The goal is to learn a transformation matrix that maps the representations in one language to those of another. To learn this transformation matrix, bilingual dictionaries are commonly utilized. Some other word-level alignment methods leverage the techniques of training monolingual word embedding but train instead on corpora containing words from both languages. These corpora can either be automatically constructed or simply corrupted. On the other hand, joint methods of learning a cross-lingual embedding space utilize parallel text as input and use the cross-lingual regularization terms in order to minimize the monolingual losses jointly.

Since they are conceptually simple and easy to use, word-level mapping-based methods of inducing a cross-lingual embedding space are by far the most popular. These methods aim to learn a mapping from independently trained monolingual spaces to a shared cross-lingual space.

### 2.2.1. Regression Method

Mikolov et al. [16] proposed the linear transformation method to learn a mapping from a monolingual to a cross-lingual embedding space. The method was inspired by the observation where semantically similar words displayed similar geometric constellations in their respective monolingual spaces upon the application of an appropriate linear transformation. This observation implies that the source monolingual embedding space can be transformed into the target space by learning a linear mapping from the former to the latter with a transformation matrix. Let $\{x_i, z_i\}_{i=1}^n$ be the set of vector representations of the translation pairs where $x_i \in \mathbb{R}^{d_1}$ is the vector representation of the $i$th word in the source embedding space and $z_i \in \mathbb{R}^{d_2}$ is the vector representation of its translation word. This

projection matrix, $W$, is learned using the $n$ most frequent words in the source embedding space and their translation in the target space as seed words. The following optimization problem is solved using Stochastic Gradient Descent (SGD):

$$min_W \sum_{i=1}^{n} ||Wx_i - z_i||^2$$

Once the translation matrix is learned using known translation pairs and their vector representations, translation for a new word in the word embedding space can be obtained by computing $z = Wx$ where $x$ is the vector representation of the word in the source language, and $z$ is the approximation of its translation in the target embedding space. Then the cosine distance is used to find the word closest to $z$ in the target language embedding space.

### 2.2.2. Orthogonal Methods

The regression method of learning a mapping from a monolingual to a cross-lingual embedding space has later been improved by constraining the transformation to be orthogonal. Xing et al. [17] motivated this constraint in order to preserve length normalization. Artetxe et al. [18], on the other hand, proposed orthogonality so that monolingual invariance is ensured. Smith et al. [19] constructed a similarity matrix

$$S = YWX^T$$

to prove that linear mapping between two semantic spaces must be orthogonal in order for it to be self-consistent. Here, X and Y are the matrices containing the word vectors in the source and target languages, respectively. Each row in these matrices represents a word vector denoted by $x$ and $y$, while each element in the similarity matrix, $S_{ij}$, computes the similarity between the $j$th source word, $x_j$ and the $i$th target word $y_i$. A second similarity matrix, $S'$, is formed such that:

$$S' = XQY^T.$$

Here Q is the transformation matrix that maps the target language back onto the source space. Similar to the first similarity matrix, an element in $S'_{ji}$ also computes the similarity between the $j$th source word and the $i$th target word. To be self-consistent, $S'$ and $S^T$ need to be equal. Since $S^T = XW^TY^T$, it can be said that $Q = W^T$, which means that the transformation matrix is orthogonal. The exact solution under the orthogonal constraint ($W^TW = I$) can be computed using Singular Value Decomposition(SVD) in linear time with respect to the size of the vocabulary, where $W$ is the transformation matrix. If $X^s$ and $X^t$ are the ordered matrices constructed from the training dictionary, the $i$th row of $X_s, Y_t$ corresponds to the word vectors in the source and target language embedding space. The SVD of $Y_t^T X_s$ is computed, resulting in $U\Sigma V^T$. Here, $U$ and $V$ are known as the left and right singular matrices, respectively, and they are both composed of orthonormal vectors, while $\Sigma$ is the diagonal matrix containing the singular values.

### 2.3. Degree of Isomorphism

Existing methods of aligning one monolingual embedding space onto another are based on the assumption that the vector spaces are approximately isomorphic. This assumption implies that the conceptual organization of the monolingual embedding spaces is language-independent. Later, researchers such as Søgaard et al. [1] and Patra et al. [4], reported poor performance and, in some cases, even failure to align two monolingual embedding spaces. From these results, they argued that the isomorphic assumption does not always stand true, and approaches based on this assumption will have significant limitations. Since poor performance was mostly reported for languages distant from English, such non-isomorphism was attributed mostly to typological differences between the languages. However, Vulić et al. [3] reported that poor BLI performance from English to a target language in a shared embedding space is correlated to the corpus size on which the

monolingual embedding spaces were trained. They also cite training duration as one of the important factors impacting the isomorphism of two vector spaces. However, they concur that typological differences among languages do impact the performance of a cross-lingual embedding space. Although not the sole deciding factor, its interplay with the corpus size affects the degree of isomorphism to a large extent. On the other hand, Ormazabal et al. [20] note that joint learning of the cross-lingual space instead of mapping independently trained vector spaces on each other results in more isomorphic embeddings.

2.3.1. Eigensimilarity

Søgaard et al. [1] propose a graph-based metric to quantify the degree of isomorphism between two monolingual embedding spaces. They treat it similarly to a graph-matching problem. Two graphs are said to match each other if they are isomorphic. For two graphs to be isomorphic, there needs to be a correspondence between the two vertex sets that preserves the adjacency as well. $G_1 = (V_1, E_1)$ is isomorphic to $G_2 = (V_2, E_2)$ if there is a bijection $\varphi : v \to v'$ such that $< x, y > \epsilon E$ if and only if $< \varphi(x), \varphi(y) > \epsilon E'$. The eigenvectors and eigenvalues of a graph determine its spectral properties. Eigenvalues represent the global properties of a graph in a compact form, and the eigenvalues of the normalized adjacency matrix of a graph are invariant under isomorphic transformations. To calculate the extent to which two monolingual embedding spaces are isomorphic, nearest-neighbor graphs are constructed for both spaces. A weak version of this requires building the nearest neighbor graphs for the most frequent words. This measure is based on the Laplacian eigenvalues of the nearest neighbor graphs [21]. Let $A_1$ and $A_2$ be the adjacency matrices of the nearest neighbor graphs constructed from the monolingual embedding spaces while the diagonal degree matrices $D_1$ and $D_2$ represent the degree of each vertex. Laplacian eigenvalues are calculated for $L_1 = D_1 - A_1$ and $L_2 = D_2 - A_2$. The smallest $k_1$ and $k_2$ are selected for the two graphs such that the sum of the largest k eigenvalues is greater than 90% of the Laplacian eigenvalues.

$$min_j \left\{ \frac{\Sigma_{i=1}^{k} \lambda_{ji}}{\Sigma_{i=1}^{n} \lambda_{ji}} > 0.9 \right\}$$

The smaller value is chosen between $k_1$ and $k_2$. This value is then used to calculate the squared difference between the top k Laplacian eigenvalue pairs of the two nearest neighbor graphs. The sum of these squared differences is taken as the Eigensimilarity, which can serve as a measure for the degree of isomorphism between two vector spaces, in this case, a pair of monolingual word embedding spaces. The higher the Eigensimilarity value between a pair of monolingual spaces, the more dissimilar they are and presumably less likely to satisfy the isomorphic assumption.

$$\Delta = \sum_{i=1}^{K} (\lambda_{1i} - \lambda_{2i})^2$$

2.3.2. Gromov–Hausdorff Distance

This metric was proposed by Patra et al. [4] in order to a priori analyze how well two monolingual spaces can be aligned under an isometric transformation. The Hausdorff distance measures the diametric distance between two metric spaces. In simpler terms, this measure computes the distance between the nearest neighbors that are the furthest apart. Let $X$ and $Y$ be the two metric spaces.

$$H(X, Y) = max\{sup_{x \epsilon X} inf_{y \epsilon Y} d(x, y), sup_{y \epsilon Y} inf_{x \epsilon X} d(x, y)\}$$

Gromov–Hausdorff distance provides a quantification of the degree of isometry between two metric spaces by minimizing this Hausdorff distance over all isometric transformations between X and Y.

$$H(X,Y) = inf_{f,g}H(f(x),g(y))$$

Here, f and g belong to a set of isometric transformations. Calculating the Gromov–Hausdorff distance between two vector spaces involves solving hard combinatorial problems. In practice, the bottleneck distance is calculated instead [22]. First, the first-order Vietoris–Rips Complex was calculated for both vector spaces. A Euclidean distance t determines the presence of an edge between two points in the Vietoris–Rips Complex. An edge exists if and only if the points are within distance t. When t = 0, there is no edge in the graph, and the vertices are all single points. As t varies and gradually increases to ∞, the points are slowly clustered together and finally form one single cluster. A persistence diagram expresses the birth and death of these clusters with the points $(t_{birth}, t_{death})$. Let f and g be two such persistence graphs while $\gamma$ is a bijection from the points of f to those of g. The bottleneck distance between f and g can be calculated as follows:

$$\beta(f,g) = inf_{gamma}(sup_{u \in f}||u - \gamma(u)||_\infty)$$

### 2.3.3. Relational Similarity

This is a comparatively straightforward measure proposed by Vulić et al. [3]. It is based on the idea that semantically equivalent words in the source and target language will have a comparable similarity distribution in their respective embedding spaces. A list is constructed with the identically spelled words shared between the source and target language space. The cosine distance is calculated for every word pair in that list on the source side. The same is repeated for the target side. The Pearson correlation coefficient is computed between the sorted list of similarity scores. Fully isomorphic embeddings will result in a coefficient value of 1, decreasing with lower degrees of isomorphism.

### *2.4. Clustering of Embedding Spaces*

### 2.4.1. Hierarchical Clustering

Hierarchical algorithms build a hierarchy of clusters. Since they output dendrograms, it is used to construct embedded classification schemes. Dendrograms are capable of simultaneously capturing the hierarchical relationship among the members at all levels of granularity. The results often depend on not only the distribution of the data but also the measure of dissimilarity and the algorithm used.

Hierarchical clustering has several advantages over partition-based clustering algorithms such as K-means. Most importantly, there is no need to specify the number of clusters to be formed. Users can traverse the dendrogram produced by hierarchical clustering to produce the desired clustering. The dendrogram structure also makes it possible to explore the entity relationships at different granularity levels [23].

Hierarchical clustering has applications in scenarios where the dataset exhibits multi-scale structure, such as phylogenetics, which aims to reconstruct the tree of biological evolution. These are the cases where partition-based clusters will not be able to discover the nested nature of the data.

### 2.4.2. Fuzzy C-Means Clustering

The task of assigning a set of objects into groups is known as clustering, where these groups are called clusters. The aim is to partition a collection of objects into clusters. Objects belonging to the same cluster are similar to each other and dissimilar from those belonging to different clusters [24]. Clustering algorithms can be broadly classified into hard and soft (fuzzy) clustering. In hard clustering, each data element can belong to only one cluster. On the other hand, fuzzy clustering allows a data element to belong to multiple clusters with a certain degree of membership. Fuzzy clustering is one of the most widely used clustering algorithms. The fuzzy set theory proposed by Zadeh used the concept of membership function to express the uncertainty of belonging [25]. Fuzzy clustering allows non-unique partitioning of a set of objects into several clusters. Instead of assigning

each data point to one cluster as performed in hard clustering, fuzzy clustering assigns membership values for each cluster. Their partial membership to all classes is expressed using a degree of membership between 0 and 1. This is particularly useful for objects that fall on the boundaries of two or more classes. Fuzzy C-Means is one of the most widely used fuzzy clustering algorithms. It was first reported by Joe Dunn for a special case (m = 2) [26]. The general case was developed and later improved by Jim Bezdek [27]. The algorithm assigns cluster membership to data points depending on their distances from the respective cluster centers. The closer a data point is to a cluster center, the greater its degree is to that cluster. The degrees of membership to different clusters add up to one for each data point. The cluster centers and the membership are updated after each iteration.

The fuzzy C-Means clustering algorithm is applied to the pairwise language distance data to find the language clusters. Clustering is an unsupervised technique. It is the purpose of this research to investigate whether real-life language families will group together to form clusters.

**3. Results**

*3.1. Hierarchical Clustering: Dendrogram*

A dendrogram demonstrates the composition of hierarchical clusters with a U-shaped link. The top of this link indicates the merging of clusters. The legs of the clusters denote which clusters have been merged using the U-link. The length of the link is a measure between the child clusters. Hierarchical clusters were produced from the pairwise distances among the languages in our dataset. The parameters were the same for all three of the distance measures, with Ward's minimization method being used as the linkage method. This method minimizes the total variance within clusters iteratively [28]. This objective function is used at every iteration as an objective function to merge new cluster pairs together. The Euclidean distance is used to measure the distance between data points.

3.1.1. Hierarchical Clustering: Eigensimilarity

Figure 2 shows the dendrogram produced from the pairwise Eigensimilarity values. It produces three clusters if cut at a distance of 400. The first cluster consists of Czech, Turkish, Romanian, Slovak, and Belarusian. The second cluster consists of Italian, Portuguese, English, Spanish, and Russian. The third cluster is made up of Azerbaijani, Galician, Greek, and Latin. The first and second clusters combine into a single cluster if the dendrogram is cut at a distance of 600. Since the two clusters are separated by a significant distance, the clustering results will be discussed with the dendrogram cut at a distance of 400, resulting in three clusters.

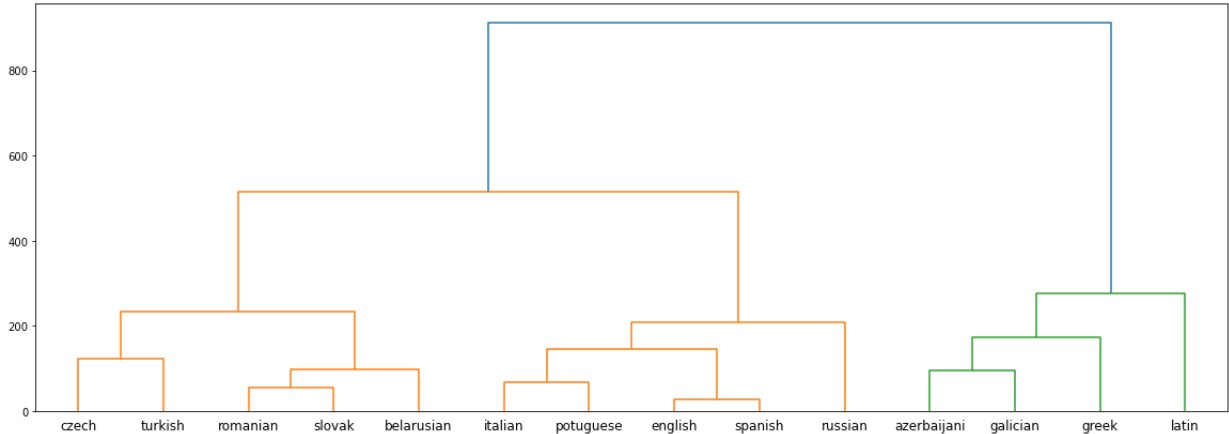

**Figure 2.** Dendrogram: Pairwise Eigensimilarity Values.

Of the languages in the first cluster, Romanian and Slovak share the lowest distance. The two of them are then clustered together to form a close-knit group of languages.

Similarly, Czech and Turkish are grouped closer together than the rest of the languages. Of the five languages forming the second cluster, English and Spanish share the lowest distance, followed by Italian and Portuguese. In the latter stage, these two pairs are grouped together at a distance below 200. Later, these four languages are grouped together with Russian at a distance of around 200. Azerbaijani and Galician are the closest languages In the third cluster. They eventually get grouped together with Greek and Latin.

**Impact of Embedding Size**

The first cluster groups low-resource Belarusian with moderate-resource Romanian, Slovak, Turkish, and resource-rich Czech. Resource-wise, the Czech space is the fifth largest, and the Belarusian space is the eleventh largest embedding space in our set of languages. Except for the sixth largest Portuguese and the tenth largest Greek, all the language spaces whose embedding sizes fall between the size range of the Czech and Belarusian spaces are part of the cluster. The languages with similar resources have lower distances among themselves in the dendrogram. The Romanian and Slovak spaces are similar in terms of available resources; they also are placed at a low distance in the dendrogram. In the dendrogram, Belarusian is placed at a low distance from Romanian and Slovak. The Czech and Turkish spaces placed closer together have a larger difference in their embedding space sizes.

The second cluster is made up of resource-rich English, Russian, Spanish, Italian, and Portuguese. Except for the fifth largest embedding space, Czech, this cluster groups together five of the six largest embedding spaces. Although English and Russian are the two largest embedding spaces, English shares the smallest distance with Spanish in the dendrogram. The Italian space is closer to the Spanish space in terms of available resources but shares a lower distance with Portuguese in the dendrodram. Russian, on the other hand, is further apart from these four languages.

The third cluster is made up of low-resource languages Latin, Galician, Azerbaijani, and moderate-resource Greek. Azerbaijani and Galician are similar in terms of available resources and are placed at a low distance in the dendrogram. The Latin space has similar resources as Azerbaijani and Galician, but moderate-resource Greek is placed at a closer distance from those two languages compared to Latin.

**Impact of Typological Similarities**

The first cluster contains languages from the Slavic (Czech, Slovak, and Belarusian), Romance (Romanian), and Turkic (Turkish) language families. Romanian, due to it being a Balkan/Eastern Romance language, has had influence from the Slavic languages. However, the hierarchical clustering algorithm applied to the pairwise Eigensimilarity values does not seem to capture the close typological relationship between Czech and Slovak. They are part of the same cluster but placed at a considerable distance in the dendrogram. The major Slavic language, Russian, is not part of the cluster, and neither is Azerbaijani—the other Turkic language in our experiment set. The second cluster contains languages from the Romance (Italian, Portuguese, Spanish), Slavic (Russian), and Germanic (English) families. English is heavily influenced by the Romance language Spanish and is also placed close to Spanish. The distance of Russian from the other languages in the cluster could be an indication of it being typologically dissimilar to them. The third language cluster is made up of Turkic (Azerbaijani), Romance (Galician), and Indo-European (Greek, Latin) languages.

From the discussion above, it seems that both embedding size and typological similarities have an impact on forming the clusters. This conclusion is supported by the inclusion of Russian with the other resource-rich languages of the experiment set. The third cluster is also formed by typologically dissimilar languages. On the other hand, low-resource Belarusian and resource-rich Czech are part of the first cluster, along with moderate-resource languages. However, it should be noted that the clear division of resources is defined by this research, and the embedding sizes of Belarusian and Czech are close to the embedding sizes of Slovak and Turkish, respectively, which are also part of the same cluster. Romanian-Slovak and Galician-Azerbaijani pairs are placed at a low distance in the dendrogram. From Figure 1, it can be seen that their embedding spaces are also of similar size. However,

languages are not completely grouped in order of their embedding resources. If that were the case, Greek and Portuguese would have been the languages to be grouped with the moderate resource languages instead of Belarusian and Czech, which indicates a level of impact from typological similarities among the languages in the cluster. Some other interesting aspects of the results include the relatively greater distance of Russian from the rest of the languages in the second cluster. In the same cluster, English and Spanish, known to be very similar languages due to the impact of French on English, are placed at a very low distance in the dendrogram.

3.1.2. Hierarchical Clustering: Gromov–Hausdorff Distance

The dendrogram produced from applying hierarchical clustering on pairwise Gromov–Hausdorff distances will be discussed in terms of the three clusters produced if the dendrogram is cut at a distance of 18. From Figure 3, it can be seen that the first of these three clusters is made up of Czech, Turkish, Russian, Spanish, Latin, Azerbaijani, Slovak, Belarusian, and Romanian. Of these, Czech-Turkish, Spanish-Latin and Slovak-Belarusian pairs share the lowest distances. Czech and Turkish are later grouped together with Russian, while Spanish-Latin and Slovak-Belarusian pairs are later grouped together with Azerbaijani and Romanian, respectively. The second cluster is closer in distance to the first one, containing spaces for the Portuguese, Greek, and English languages. The third cluster is made up of just two languages, Italian and Galician.

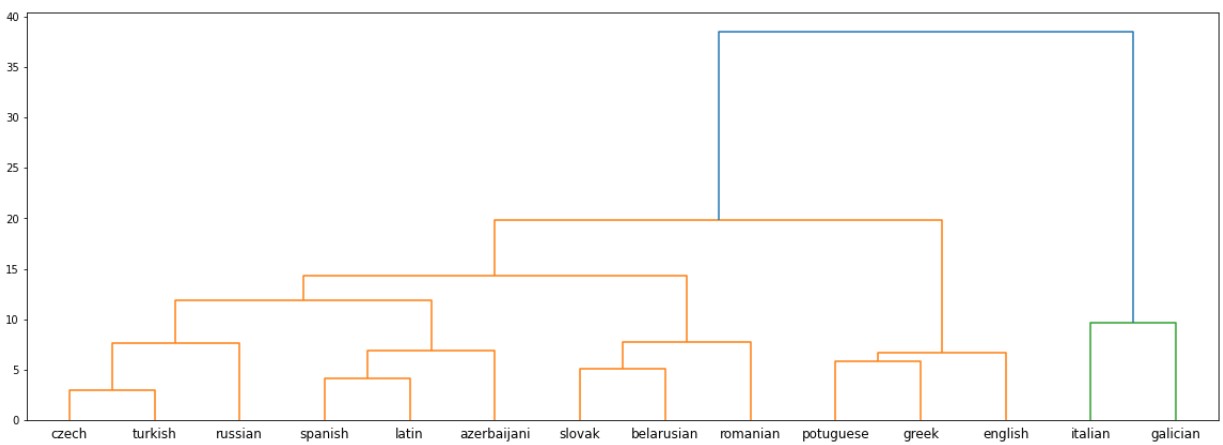

**Figure 3.** Dendrogram: Pairwise Gromov–Hausdorff Distance Values.

**Impact of Embedding Size**

The first major cluster in the dendrogram is made up of low-resource (Latin, Azerbaijani, and Belarusian), moderate-resource (Slovak, Romanian, and Turkish), and resource-rich (Czech and Russian) languages. If the dendrogram is cut at a distance of 10 to closely examine the smaller clusters that join to form the first cluster, it shows some possible instances of grouping languages with similar resources. Although the resource classifies Turkish as a moderate-resource language, the disparity in resources between Turkish and Czech is relatively lower than that between Czech and Russian. Similarly, Belarusian, Slovak, and Romanian are similar in terms of available resources. The only other language with a similar embedding size that is missing from this cluster is Greek. Latin and Azerbaijani are low-resource languages, but the embedding size of Spanish is greater than both of them. The second cluster has two resource-rich languages—Portuguese and English, with moderate-resource Greek. The third cluster groups together low-resource Galician with resource-rich Italian.

**Impact of Typological Similarity**

The first major cluster in the dendrogram clusters languages from the Romance (Romanian and Spanish), Slavic (Belarusian, Slovak, Czech, and Russian), Turkic (Azerbaijani and Turkish) language families, along with Latin. A closer look, however, does not consistently

reveal a closer distance that correlates with the language families. If the dendrogram is cut at a distance of 10 to examine the languages in the first cluster at a more granular level, it can be seen that Czech, Turkish, and Russian are placed close together. Czech and Russian are Western and Eastern Slavic languages, respectively, while Turkish belongs to the Turkic language family. Another smaller cluster places the Romance language Spanish with Latin, and the Turkic language Azerbaijani close together. It can be argued that as the source of Romance languages, the clustering of Latin with Spanish makes sense. These smaller clusters are joined at a distance of 12. The third of these clusters places Slavic languages Belarusian and Slovak with Slavic-influenced Romance language Romanian. The second larger cluster formed at a distance of 18 groups Romance language Portuguese with Romance-influenced Germanic language English together with Greek. Greek forms its own independent branch in the Indo-European language tree. The third and final cluster groups two Romance languages, Galician and Italian. Although both these languages are Romance languages, Galician is known to be typologically more similar to Portuguese and Spanish than Italian.

From the discussion above, it seems unlikely that embedding space size has a significant impact on forming the clusters from pairwise Gromov–Hausdorff distances. Slovak, Belarusian, and Romanian being placed close together on the dendrogram could result from both their embedding sizes being similar and their typological relatedness. The dendrogram shows several instances of languages of disparate resources being placed in close proximity. Notable among them is the close distance between Latin-Spanish, Portuguese-Greek, and Italian-Galician pairs. There are also instances of languages with similar resources being grouped together, such as the Czech-Turkish and, as mentioned before, Belarusian-Slovak-Romanian.

### 3.1.3. Hierarchical Clustering: Relational Similarity

The dendrogram in Figure 4 is produced from pairwise Relational Similarities and shows three clusters if cut at a distance of 1.25. The first cluster is made up of Turkish, Romanian, Galician, Czech, Slovak, Greek, Latin, and Azerbaijani. Of these languages, Czech and Slovak share the smallest distance. They are followed by Turkish and Romanian, which are later grouped together with Galician. Similarly, Greek and Latin share a closer distance within this larger cluster. Finally, these languages are grouped together with Azerbaijani, which has a greater distance from the other languages in the clusters than they have among themselves. The second cluster is made up of only two languages, Russian and Belarusian. The third cluster is made up of Spanish, Portuguese, Italian, and English. Of these, Spanish and Portuguese have a low distance, which is then followed by Italian and then English.

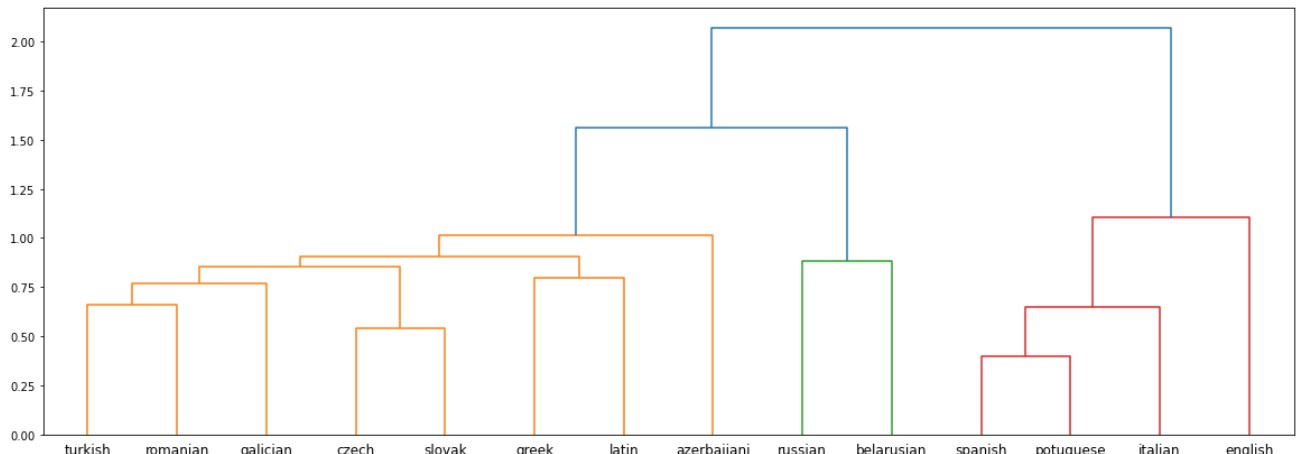

**Figure 4.** Dendrogram: Pairwise Relational Similarity Values.

**Impact of Embedding Size**

The first cluster groups together low-resource (Latin, Azerbaijani, and Galician), moderate-resource (Greek, Slovak, Romanian, and Turkish) with resource-rich Czech. It should be noted that with the exception of Belarusian and Portuguese, these languages are among the first ten embedding spaces if they were sorted in increasing order of their sizes. Hence, a closer look at the cluster might reveal more information. If the dendrogram is cut at a lower distance, it produces four clusters from the languages in the first cluster. The first of them contains the Turkish, Romanian, and Galician spaces. Turkish and Romanian are both moderate-resource languages with a very similar number of words making up the embedding spaces, but Galician is a low-resource language. The second of these smaller clusters contains the Czech and Slovak spaces. Czech is a resource-rich language, while Slovak is a moderate-resource language in our language set. The third of these clusters is made up of Greek and Latin, which are moderate-resource and low-resource, respectively. The fourth cluster is made of a single language, Azerbaijani. The second major cluster contains the language spaces of Russian and Belarusian. The Russian space is the second largest embedding space in our language space, while the Belarusian space is the fourth smallest one. The final and third major cluster is made up of resource-rich spaces of Spanish, Portuguese, Italian, and English. These are among the six largest embedding spaces in our language space, the remaining two being Russian and Czech.

**Impact of Typological Similarity**

The first cluster groups together languages from the Slavic (Czech and Slovak), Turkic (Turkish and Azerbaijani), and Romance (Romanian and Galician), along with Indo-European Greek and Latin. Taking a closer look at the different clusters if the dendrogram is cut at a lower distance reveals smaller clusters where Romanian and Turkish are grouped close together and then clustered with Galician at an increased distance. Romanian has had some influence from Turkish, while Galician and Romanian are both Romance languages. More interesting is the result where two Slavic languages, specifically Western Slavic languages, are grouped together at a low distance. Greek and Latin are grouped together with a low distance between them. Of the languages in the experiment set, these two can be considered independent. While Latin is the language from which all Romance languages were generated, modern Romance languages have diverged from Latin. Greek forms its own independent branch in the Indo-European language family. These two languages being grouped together is also an interesting aspect of the result. The second major cluster is formed with Eastern Slavic languages Belarusian and Russian. The third and final cluster is formed with Romance languages Spanish, Portuguese, Italian, and Spanish-influenced Germanic language English. A closer look at the cluster at a lower distance reveals that Spanish and Portuguese are placed at a very low distance, reflecting their typological similarity since they both belong to the Ibero-Romance sub-family. These two languages are then grouped with Italian at a slightly higher distance. Italian, although a Romance language, belongs to a different branch of the Romance family tree. Finally, English is grouped with these three Romance languages at an even greater distance. This correlates with the fact that although English has been heavily influenced by the Romance languages, especially French, typologically, it is a Germanic language.

From the discussion above, it seems that the clusters produced by the hierarchical clustering algorithm from pairwise Relational Similarity values are impacted more by typological similarities than the size of the embedding space. Some of the results that provide strong support for this include the grouping together of Czech-Slovak, Spanish-Portuguese, and Russian-Belarusian pairs. Czech-Slovak and Russian-Belarusian pairs are disparate in terms of their embedding sizes. The cluster that groups Spanish, Portuguese, Italian, and English reflects in its inter-lingual distances the relationships that these languages share among themselves. On the other hand, the grouping of languages that may be influenced by the language pair's embedding size is the Romanian-Turkish pair. Both languages have a similar number of words in their respective embedding spaces. However, they are also joined by Galician at a slightly higher distance, which is a decidedly low-resource language.

Another reason for this grouping may be that Romanian has had some Turkish influence over the years, which may be embedded in their language spaces. Further, both Galician and Romanian are Romance languages. Romanian has distinct features due to it being a Balkan/Eastern Romance language. Galician is typologically similar to Spanish and Portuguese. However, its lack of resources compared to these Ibero-Romance languages may have resulted in a divergence that has caused it to be closer in the dendrogram to Romanian than the rest of the Romance languages. The language whose positioning in the dendrogram cannot be clearly explained is Azerbaijani. Although it is part of the same larger cluster that Turkish is part of, the distance between these two Turkic languages is quite significant, and it is grouped together with the rest of the languages in the first cluster at the same distance.

### 3.2. Fuzzy C-Means Clustering Algorithm

Based on the results in Section 3.1, the number of clusters was set to three for the Fuzzy C-Means clustering algorithm.

### 3.2.1. FCM: Eigensimilarity

Table 1 shows the results from the FCM clustering algorithm applied on pairwise Eigensimilarity values with the languages divided into three clusters. Latin, Galician, Azerbaijani, and Greek form Cluster 0. Of these, Galician has the highest degree of membership in its assigned cluster. Latin, Greek, and Azerbaijani have memberships of 0.12, 0.15, and 0.20 to Cluster 2 as well. Cluster 1 contains the languages Portuguese, Italian, Spanish, Russian, and English. Of these, Italian and Russian have 0.21 and 0.12 memberships to Cluster 2 while having 0.74 and 0.82 membership to their assigned cluster label. The third cluster, with label 2, is made up of languages Belarusian, Slovak, Romanian, Turkish and Czech. Of these, Turkish and Czech display a considerable degree of membership to Cluster 1 with 0.33 and 0.42 membership, respectively, while they belong to their assigned cluster with 0.58 and 0.52 membership, respectively.

**Table 1.** FCM Results for Pairwise Eigensimilarity Values (*n* = 3).

| Language | Cluster Label | 0 | 1 | 2 |
|---|---|---|---|---|
| Latin | 0 | 0.816125 | 0.066704 | 0.11717 |
| Galician | 0 | 0.947083 | 0.015154 | 0.037764 |
| Azerbaijani | 0 | 0.74006 | 0.061298 | 0.198643 |
| Greek | 0 | 0.798175 | 0.052795 | 0.14903 |
| Portuguese | 1 | 0.010361 | 0.951072 | 0.038567 |
| Italian | 1 | 0.04371 | 0.742565 | 0.213725 |
| Spanish | 1 | 0.00419 | 0.980757 | 0.015053 |
| Russian | 1 | 0.058801 | 0.816661 | 0.124538 |
| English | 1 | 0.006692 | 0.971533 | 0.021775 |
| Belarusian | 2 | 0.075486 | 0.063486 | 0.861028 |
| Slovak | 2 | 0.021567 | 0.022205 | 0.956228 |
| Romanian | 2 | 0.014296 | 0.020679 | 0.965025 |
| Turkish | 2 | 0.089349 | 0.330027 | 0.580625 |
| Czech | 2 | 0.054219 | 0.417626 | 0.528155 |

**Impact of Typological Similarity**

Cluster 0 contains a Romance language, Galician, a Turkic language, Azerbaijani, along with general Indo-European languages Latin and Greek. Cluster 1 contains languages from the Romance (Portuguese, Italian, and Spanish), Slavic (Russian), and Germanic (English) language families. Cluster 2 contains languages from Slavic (Belarusian, Slovak, and Czech), Romance (Romanian), and Turkic (Turkish) language families. It makes sense for English to be clustered with the Romance languages due to their heavy influence on the language. Russian, which is a Slavic language clustered with Romance languages with 0.82 membership. It has 0.12 membership to Cluster 2, which contains the other Slavic

languages of the language set. Romanian is assigned to Cluster 2 with a high degree of membership (0.97). Although a Romance language, it has had influences from the Slavic languages. Czech is assigned to Cluster 2 but has a considerable degree of membership to Cluster 1 (0.42). Turkish, too, is assigned to Cluster 2 with 0.58 membership, while it has 0.33 membership to Cluster 1.

**Impact of Embedding Size**

The languages in Cluster 0 are the first, second, third, and fifth lowest resource languages in our experiment set. On the other hand, Cluster 1 is made up of five of the six highest resource languages in our experiment set. Belarusian, which is among the low-resource languages, and Czech, which is among the resource-rich languages, are clustered together with Slovak, Romanian, and Turkish, which are moderate resource languages in our language set. Czech is assigned to Cluster 2 with 0.53 membership, while it has around 0.42 membership to Cluster 1, which contains the other high-resource languages in the language set. A similar trend is not visible in Belarusian. It belongs to Cluster 2 with 0.86 membership, while it belongs to Cluster 0 and Cluster 1 with around 0.08 and 0.06 membership.

The embedding spaces for the languages in Cluster 0 are the first, second, third, and fifth smallest languages in the dataset. Neither of these four languages is etymologically close to each other. However, Galician, being a Romance language, may have some similarities with Latin. Of these, Portuguese, Italian, and Spanish are major Romance languages, while English is also heavily influenced by the Romance family of languages. Russian, however, is a Slavic language. The embedding spaces for these languages are the first, second, third, fourth, and sixth largest in our dataset. Among these languages, Belarusian, Czech, and Slovak are Slavic languages. Romanian, even though a Romance language, has strong Slavic influences. Turkish is from the Turkic family of languages. The inclusion of Belarusian and Czech in Cluster 2 may have resulted from the typological similarity they share with Slovak. However, the inclusion of Russian in Cluster 1 with other typologically distant but similarly resourced languages, as well as the high degree of membership of Czech to Cluster 1 indicate that the embedding size has a marginally stronger effect.

3.2.2. FCM: Gromov–Hausdorff Distance

The results of the FCM clustering algorithm can be seen in Table 2. In this case, one of the three clusters is identical to the one formed in the previous iteration of Fuzzy C-Means clustering on the pairwise Gromov–Hausdorff distances. This cluster, labeled 1 here, consists of Italian and Galician. Galician has 0.05 membership to Cluster 0 and 0.08 membership to Cluster 2. Italian, on the other hand, has 0.03, 0.93, and 0.04 membership to Clusters 0, 1, and 2, respectively. Of the other two clusters, Cluster 0 is made up of Azerbaijani, Belarusian, Slovak, Romanian, Turkish, Czech, and Russian. Of these languages, Slovak and Czech show a similar distribution of membership across the Clusters. A similar trend can be seen in the distribution of membership for Turkish-Romanian and Azerbaijani-Russian pairs. Finally, Cluster 2 is made up of Latin, Greek, Spanish, Portuguese, and English, where Latin and Spanish show a similar distribution of membership across the three clusters, while Greek, Portuguese, and English also have similar degrees of cluster membership.

**Impact of Embedding Size**

Cluster 0 is made up of low-resource (Azerbaijani and Belarusian), moderate-resource (Slovak, Romanian, and Turkish), and resource-rich (Czech and Russian) languages. Cluster 1 is made up of two languages, with one of them being a low-resource language while the other is a resource-rich language. Cluster 2 contains low-resource (Latin), moderate-resource (Greek), and resource-rich (Portuguese, Spanish, and English) languages. In Cluster 0, low-resource Azerbaijani and resource-rich Russian have the same distribution of membership across clusters. Romanian and Turkish also show a similar distribution, but they are comparable in terms of the resources their respective embedding spaces were

trained on. Slovak and Czech are disparate in their embedding sizes, with Slovak being a moderate-resource language and Czech a resource-rich one. These two languages also have a similar membership distribution across clusters. Similarly, in Cluster 2, low-resource Latin and high-resource Spanish have similar membership distribution. Greek, a moderate resource language, is also similar in distribution to resource-rich Portuguese and English.

**Table 2.** FCM Results for Pairwise Gromov–Hausdorff Values (*n* = 3).

| Language | Cluster Label | 0 | 1 | 2 |
|---|---|---|---|---|
| Azerbaijani | 0 | 0.495611 | 0.049662 | 0.454728 |
| Belarusian | 0 | 0.901068 | 0.012263 | 0.086669 |
| Slovak | 0 | 0.852028 | 0.023885 | 0.124087 |
| Romanian | 0 | 0.689999 | 0.070371 | 0.239631 |
| Turkish | 0 | 0.661249 | 0.024543 | 0.314208 |
| Czech | 0 | 0.80863 | 0.01881 | 0.172561 |
| Russian | 0 | 0.547999 | 0.056037 | 0.395964 |
| Galician | 1 | 0.051071 | 0.867585 | 0.081343 |
| Italian | 1 | 0.02978 | 0.926727 | 0.043492 |
| Latin | 2 | 0.341101 | 0.03313 | 0.625769 |
| Greek | 2 | 0.129552 | 0.035976 | 0.834472 |
| Portuguese | 2 | 0.160704 | 0.074351 | 0.764946 |
| Spanish | 2 | 0.323441 | 0.032605 | 0.643954 |
| English | 2 | 0.157306 | 0.090714 | 0.75198 |

**Impact of Typological Similarity** Cluster 0 consists of languages from the Turkic (Azerbaijani and Turkish), Slavic (Belarusian, Slovak, Czech, and Russian), and Romance (Romanian) family of languages. However, the membership distribution of Azerbaijani and Turkish is not similar. Rather Azerbaijani and Russian show a similar membership distribution as do Romanian and Turkish. There have been some influences of Turkish on Romanian over the years, which may have contributed to the similarity in distribution. Slovak and Czech, two typologically similar languages, also display similar membership distribution, which is shared also by Belarusian, yet another Slavic language. Cluster 1 has two Romance languages, Galician and Italian. Cluster 2 is made up of languages from the Romance family of languages (Portuguese and Spanish) with the Romance-influenced Germanic language English. The cluster also contains Latin and Greek. Of these languages, Greek, Portuguese, and English have similar distributions, while Latin and Spanish share similar distributions.

From the discussions above, it is unlikely that embedding size is playing a role in the clustering of languages on their pairwise Gromov–Hausdorff distances using the Fuzzy C-Means clustering algorithm. On the other hand, the distance measure and the clustering algorithm used seem to be encoding some typological information. Slavic languages Belarusian, Slovak, Czech, and Russian are all in the same cluster, with Belarusian, Czech, and Slovak showing a similar distribution of cluster membership. Azerbaijani, Turkish, and Romanian are also part of the same cluster. Turkish and Azerbaijani are typologically related, while Romanian has had influences from both the Slavic languages as well as Turkish. On the other hand, the Romance languages in our language set are divided into two different clusters, with Galician and Italian forming Cluster 1 and Spanish and Portuguese forming Cluster 2 together with English, Latin, and Greek. Although Galician is known to be typologically closer to Portuguese and Spanish, its clustering with a Romance language is still significant because it is a severely low-resource language, which is known to be one of the factors affecting the isomorphism between vector spaces. The clustering of English together with major Romance languages is consistent with its real-world influence from the Romance languages.

### 3.2.3. FCM: Relational Similarity

Table 3 shows the three clusters formed from the pairwise Relational Similarity values. Galician, Azerbaijani, Slovak, Romanian, Turkish, and Czech make up Cluster 0. Although, broadly, all five languages have a similar degree of membership, hence clustered under the same cluster label, they vary in their membership to Clusters 1 and 2. Galician and Romanian are distributed almost evenly between Clusters 1 and 2. On the other hand, Azerbaijani, Slovak, Turkish, and Czech have a comparatively higher degree of membership to Cluster 2 compared to Cluster 1. Portuguese, Italian, Spanish, and English make up Cluster 1. Of these, Portuguese, Italian, and Spanish are more similar in their membership distribution. Their membership to the three clusters ranges between 0.11–0.16, 0.72–0.81, and 0.08–0.12, respectively. On the other hand, English is distributed with a 0.25–0.53–0.22 membership across the three clusters. Latin, Belarusian, Greek, and Russian make up Cluster 2. Latin is slightly different in membership distribution across the three clusters. Of these, Belarusian, Russian, and Greek diverge from Latin with respect to their membership to Clusters 0 and 2. Latin belongs to Clusters 0 and 2 with 0.40 and 0.45 membership. The other three languages' memberships to Clusters 0 and 2 fall between 0.30–0.34 and 0.49–0.55. The memberships to Cluster 1 for these languages range between 0.10–0.21.

**Table 3.** FCM Results for Pairwise Relational Similarity Values ($n = 3$).

| Language | Cluster Label | 0 | 1 | 2 |
|----------|---------------|---|---|---|
| Galician | 0 | 0.441000 | 0.266008 | 0.292991 |
| Azerbaijani | 0 | 0.457129 | 0.224005 | 0.318866 |
| Slovak | 0 | 0.496797 | 0.127349 | 0.375854 |
| Romanian | 0 | 0.487956 | 0.239847 | 0.272196 |
| Turkish | 0 | 0.547817 | 0.167559 | 0.284624 |
| Czech | 0 | 0.515586 | 0.178321 | 0.306093 |
| Portuguese | 1 | 0.158451 | 0.732514 | 0.109034 |
| Italian | 1 | 0.155662 | 0.723955 | 0.120383 |
| Spanish | 1 | 0.110976 | 0.810494 | 0.078530 |
| English | 1 | 0.247609 | 0.535148 | 0.217243 |
| Latin | 2 | 0.402406 | 0.152284 | 0.445310 |
| Belarusian | 2 | 0.301698 | 0.154456 | 0.543846 |
| Greek | 2 | 0.341317 | 0.102445 | 0.556238 |
| Russian | 2 | 0.298074 | 0.207581 | 0.494345 |

#### Impact of Embedding Size

Of the five languages making up Cluster 1, Galician and Azerbaijani are low-resource languages; Slovak, Romanian, and Turkish are moderate-resource languages, while Czech is a resource-rich language. Galician and Azerbaijani do show a similar distribution of membership across the three clusters. Slovak, Romanian, and Turkish belong to Cluster 0, with membership ranging from 0.49–0.54. Their degrees of membership to Cluster 1 fall between a wider range (0.13–0.24). Among these languages, Slovak has the lowest degree of membership at 0.13, followed by Turkish at 0.18 and Romanian at 0.24. Their membership degrees to Cluster 2 also have a similar range (0.27–0.38). Slovak has the highest degree of membership to Cluster 2 at 0.38, while Turkish and Romanian belong to Cluster 2 with a membership of 0.28 and 0.27, respectively. Czech belongs to the three clusters with 0.51, 0.18, and 0.31 membership.

The languages making up Cluster 1—Portuguese, Italian, Spanish, and English—are all resource-rich languages. As mentioned above, the membership distribution is slightly different from the other three languages. The English embedding space is, however, significantly larger in size than the other three, with around 2.5 million words. The Spanish space, the third largest in our set of languages, contains 985,667 words.

Cluster 2 is made up of low-resource Latin and Belarusian, moderate-resource Greek, and resource-rich Russian. Latin has the lowest membership to its assigned cluster and has a significant membership to Cluster 0. Of the remaining languages, Belarusian and Greek

have a higher degree of membership to Cluster 0, compared to Russian, which has a more evenly distributed membership degree to Clusters 0 and 1.

**Impact of Typological Similarity** Cluster 0 is made up of languages from Romance (Galician and Romanian), Turkic (Azerbaijani and Turkish), and Slavic (Slovak and Czech) languages. As mentioned while discussing the impact of embedding size in forming the clusters, Galician and Romanian have a similar distribution. However, their membership is evenly distributed between Clusters 1 and 2, clusters that contain the major Romance and Slavic languages, respectively. The Slavic languages, as well as the Turkic languages, are similar in their membership distributions. Romance languages Portuguese, Italian, and Spanish are clustered with the Romance-influenced Germanic language English. On the other hand, Cluster 2 is made up of Slavic languages Belarusian and Russian, along with Latin and Greek.

From the discussion above, it seems that typological similarities impact the language clusters more than their embedding sizes. Languages with disparate resources are clustered together, while languages belonging to the same language family are clustered together. Although Galician belonging to the same cluster as Romanian may not be an indication of both being Romance languages, as Azerbaijani is also part of the cluster, and both Azerbaijani and Galician are low-resource languages. A similar explanation can be provided for Belarusian being clustered with Russian, Greek, and Latin, where Belarusian and Russian are Slavic languages and Latin and Belarusian are low-resource languages. Although the similar membership distributions of Romanian and Galician make the case stronger for it being an impact of their typological similarities. A similar phenomenon is also noticed in the membership distributions of Belarusian and Russian but less so in the membership distributions of Azerbaijani and Turkish—two Trukic languages. Interestingly, Czech and Slovak are clustered together in Cluster 0, while Russian and Belarusian are clustered together in Cluster 2. This could possibly be an indication of the two pairs of languages belonging to different branches of the Slavic family tree. Of the languages forming Cluster 1, the Romance languages show similar degrees of membership to the three clusters, while English shows a different distribution. This could be both an indication of English being relatively richer in resource than the other three resource-rich languages and English being a Germanic language, albeit with heavy Romance influences.

## 4. Discussion

Cross-lingual word embeddings have enabled the transferring of knowledge from one language to another. In particular, such transfers can greatly benefit low-resource languages on the internet. However, cross-lingual word embeddings for low-resource languages perform poorly compared to resource-rich languages. Such unsatisfactory performance has often been attributed to the weak isomorphic assumption on which the mapping-based methods of inducing cross-lingual embeddings heavily rely. The goal of this paper was to investigate the comparative impact of corpus size and typological relationship among languages on the degree of isomorphism between their respective embedding spaces. The goals are motivated first by the scholarly interest of finding out the extent to which the independently trained word embeddings capture the linguistic properties that define the inter-lingual relationships among a set of languages. Secondly, from an engineering point of view, such findings can help improve the performance of low-resource languages in a cross-lingual embedding space. Previous works have utilized the concept of isomorphism measure to analyze the poor results of low-resource languages in a cross-lingual embedding space [1,3]. This research experiments with different degrees of isomorphism between the monolingual embedding spaces in order to determine their viability in encoding the real-world linguistic relationship among languages. Clusters formed from pairwise degrees of isomorphism can help identify related higher-resource languages or low-resource languages in a set of languages. Such clusters of languages can help induce better-performing cross-lingual word embeddings.

This research presented the results of two clustering algorithms applied to the pairwise degrees of isomorphism among the fourteen chosen languages. The algorithms applied are the hierarchical clustering algorithm and Fuzzy C-Means. Hierarchical clustering algorithm was chosen in order to study the structure of the clusters at a granular level. The dendrogram produced as an output of the clustering algorithm captured the hierarchical relationship among the embedding spaces in our language space. Fuzzy C-Means is a popular Fuzzy Clustering algorithm where each member belongs to the clusters formed with a degree of membership instead of being uniquely partitioned into distinct clusters.

The clustering algorithms produced similar clusters with Eigensimilarity used as the measure of isomorphism between a pair of monolingual embedding spaces when the number of clusters was set to three. These three clusters were formed with the following groups of languages: Latin-Galician-Azerbaijani-Greek, Belarusian-Slovak-Romanian-Turkish-Czech, and Portuguese-Italian-Spanish-Russian-English. The impact of the embedding space size seems to be the dominant factor in forming these clusters.

The clusters formed by the clustering algorithms based on the pairwise Relational Similarity values are also similar to an extent. The set of languages grouped together by the hierarchical clustering algorithm Latin-Galician-Azerbaijani-Greek-Slovak-Romanian-Turkish-Czech, Belarusian-Russian, and Portuguese-Italian-Spanish-English. Inspecting the clusters at a more granular level in the dendrogram produced from the hierarchical clustering algorithm revealed a closer distance between typologically similar languages, such as Czech-Slovak, Belarusian-Russian, and Portuguese-Spanish pairs. The dendrogram also placed the embeddings for the Latin and Greek languages in close proximity. FCM algorithm, however, places Greek and Latin together with Belarusian and Russian.

The clusters produced by the algorithms applied on the pairwise Gromov–Hausdorff distances are markedly different from each other. Although, in both cases, Italian and Galician form one of the clusters. The other two clusters formed by the hierarchical clustering algorithm are Latin-Azerbaijani-Belarusian-Slovak-Romanian-Turkish-Czech-Russian and Greek-English-Portuguese. The clusters formed by the FCM algorithm are Galician-Italian, Latin-Greek-Portuguese-Spanish-English, and Azerbaijani-Belarusian-Slovak-Romanian-Turkish-Czech-Russian. Although the languages clustered together are typologically similar, they do not reveal a low distance between closely related languages at a granular level.

From the results discussed above, it is clear that of the three measures of isomorphism studied, it is Relational Similarity that best captures the typological relationship among the chosen set of languages. Both the clustering algorithms applied to the pairwise Relational Similarity values group typologically related languages together. At a granular level, the hierarchical clustering algorithm places related languages closer together in the dendrogram. In the clusters produced by the Fuzzy C-Means clustering algorithm, the related languages are clustered together. However, closely related languages do not always show a similar distribution of membership, as was expected.

On the other hand, Eigensimilarity as a measure of isomorphism does not seem to encode the typological relationships among the languages. Rather, the clusters formed from the pairwise Eigensimilarity values suggest that the relative corpus size of the embedding spaces is the deciding factor. The Gromov–Hausdorff distance does not encode the information regarding the relative corpus size as the clusters group together languages with disparate resources. The clusters formed reflect some of the real-world relationships among the languages, but the results are not as strong as those achieved using the pairwise Relational Similarity values.

**Author Contributions:** Conceptualization, K.B. and A.R.; methodology, K.B.; software, K.B.; validation, K.B. and A.R.; investigation, K.B.; resources, K.B.; data curation, K.B.; writing—original draft preparation, K.B.; writing—review and editing, A.R.; visualization, K.B.; supervision, A.R.; project administration, A.R. All authors have read and agreed to the published version of the manuscript.

**Funding:** This research received no external funding.

**Data Availability Statement:** The data presented in this study are openly available in reference [6].

**Conflicts of Interest:** The authors declare no conflict of interest.

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
