# Peer review of "Clustering of Monolingual Embedding Spaces"

_digital, doi:10.3390/digital3010004_

Round 1

Reviewer 1 Report

Fascinating article and also current. Artificial intelligence is increasingly used in various fields, and research in the field of borrowing words between languages is presented here. Methodologically, the article is well thought out, and the use of methods is well justified. However, the lack of sources in the first chapter is surprising, as one would expect claims about the relatedness of languages, etc. supported by various sources. In this way, the interested reader could deepen his understanding of specific questions that would arise. In any case, congratulations on an interesting paper.

Author Response

Response submitted as in the uploaded PDF file

Reviewer 2 Report

Clustering of Monolingual Embedding Spaces

Kowshik Bhowmik and Anca Ralescu

In this paper, the autthor investigated the comparative impact of typological relationship and corpus size on the isomorphism between monolingual embedding spaces. To that end, two clustering algorithms were applied to three sets of pairwise degrees of isomorphism. It is also the goal of the paper to determine the combination of the isomorphism measure and clustering algorithm that best captures the typological relationship among the chosen set of languages. Of the three measures investigated, Relational Similarity seemed to capture best the typological information of the languages encoded in their respective embedding spaces.  The issues with this paper are as follows:

The main issue with this paper is that the research gaps are not properly identified and presented. The key motivation behind this research is also not clear. I noticed in the last paragraph, authors’ highlighted motivations for this work. However, these are not motivating enough to the readers. 

The conclusion can be strongly revised. This reviewer strongly suggests improving the flow of the conclusion section. Start with a brief explanation of the paper's goal (like the abstract), but make sure that the conclusion is different from the abstract. Provide the main findings/claims. Explain the numerical findings of the simulations. Clearly explain what the significant findings are and why your paper is really important. 

The poor presentation and linguistic quality are the major concerns with this work, making often hard to follow. 

Please check all the mathematical expressions very carefully.

Expand abbreviations in the first place of occurrence.

Author Response

Response submitted as a PDF file.

Reviewer 3 Report

In this reviewed paper, the authors investigate the comparative impact of typological relationship and corpus size on the isomorphism between monolingual embedding spaces. I think that some of the content of this paper is weak. However, it can be improved as follows:
1. Please focus on the generally significant aspects of this paper.  2. The abstract part is not complete, please revise and add the strong advantage of this paper in the abstract part.
2. In this version, I think that the results of this paper are weak. Please improve the presentation and extend it significantly.
3. Please check for typos and grammar errors. For example Line 35 :  1.1. fastText Word Vectors    Line 36 : the full stop. 4. Please verify  Figure 1. I don't understand le6.  

Author Response

Response submitted as a PDF file.

Round 2

Reviewer 3 Report

The revised paper is improved. The quality of this paper can be published.